# Renal Cell Carcinoma with Sarcomatoid Features: Finally New Therapeutic Hope?

**DOI:** 10.3390/cancers11030422

**Published:** 2019-03-25

**Authors:** Renate Pichler, Eva Compérat, Tobias Klatte, Martin Pichler, Wolfgang Loidl, Lukas Lusuardi, Manuela Schmidinger

**Affiliations:** 1Department of Urology, Medical University Innsbruck, A-6020 Innsbruck, Austria; 2Department of Pathology, Hôspital Tenon, HUEP, Sorbonne University, 75005 Paris, France; eva.comperat@aphp.fr; 3Department of Urology, Royal Bournemouth Hospital, Bournemouth BH7 7DW, UK; Tobias.Klatte@gmx.de; 4Division of Clinical Oncology, Internal Medicine, Medical University of Graz, 8036 Graz, Austria; Martin.Pichler@medunigraz.at; 5Division of Cancer Medicine, MD Anderson Cancer Center, Houston, TX 77030, USA; 6Department of Urology, St Vincent’s Hospital of Linz, 4010 Linz, Austria; Wolfgang.Loidl@ordensklinikum.at; 7Department of Urology & Andrology, Paracelsus Medical University Salzburg, 5020 Salzburg, Austria; l.lusuardi@salk.at; 8Clinical Division of Oncology, Department of Medicine I & Comprehensive Cancer Center, Medical University of Vienna, 1090 Vienna, Austria; Manuela.Schmidinger@meduniwien.ac.at

**Keywords:** sarcomatoid, RCC, immunotherapy, checkpoint inhibitors, survival, PD-L1

## Abstract

Renal cell carcinoma (RCC) with sarcomatoid differentiation belongs to the most aggressive clinicopathologic phenotypes of RCC. It is characterized by a high propensity for primary metastasis and limited therapeutic options due to its relative resistance to established systemic targeted therapy. Most trials report on a poor median overall survival of 5 to 12 months. Sarcomatoid RCC can show the typical features of epithelial-mesenchymal transition (EMT) and may contain epithelial and mesenchymal features on both the morphological and immunhistochemical level. On the molecular level, next-generation sequencing confirmed differences in driver mutations between sarcomatoid RCC and non-sarcomatoid RCC. In contrast, mutational profiles within the epithelial and sarcomatoid components of sarcomatoid RCC were shown to be identical, with TP53 being the most frequently altered gene. These data suggest that both epithelial and sarcomatoid components of RCC originate from the same progenitor cell, segregating primarily according to the underlying histologic epithelial subtype of RCC (clear cell, papillary or chromophobe). Current studies have shown that sarcomatoid RCC express programmed death 1 (PD-1) and its ligand (PD-L1) at a much higher level than non-sarcomatoid RCC, suggesting that blockade of the PD-1/PD-L1 axis may be an attractive new therapeutic strategy. Preliminary results of clinical trials evaluating checkpoint inhibitors in patients with sarcomatoid RCC showed encouraging survival data and objective response and complete response rates of up to 62% and 18%, respectively. These findings may establish a new standard of care in the management of patients with sarcomatoid RCC.

## 1. Introduction

Renal cell carcinoma (RCC) with sarcomatoid differentiation (sRCC) is a highly aggressive form of RCC. Histologically, sRCC shows loss of characteristic epithelial components and contains features such as spindle cells, high cellularity, and cellular atypia. These features are found in 5–8% of clear-cell RCC (ccRCC), 8–9% of chromophobe RCC, and 2–3% of papillary RCC [1,2,3,4]. About 75% of patients with sRCC present with metastatic disease [5,6] and outcomes are generally modest. Therapeutic strategies include vascular endothelial growth factor (VEGF)-targeted monotherapy [7], and combined strategies with sunitinib plus gemcitabine [8] or gemcitabine plus doxorubicin [9]. The majority of studies report on a poor median overall survival, ranging from 5 to 12 months [7,8,9].

## 2. The sRCC

The sRCC is not a distinct morphogenetic subtype of RCC [3,10]. It originates from epithelial-mesenchymal transition (EMT), and therefore contains both epithelial (carcinoma) and mesenchymal (sarcomatoid) features on both the morphological and immunhistochemical level [11], which is distinctive from primary sarcoma of the kidney [3,10]. The presence of even a small component of sarcomatoid differentiation was shown to independently predict poor survival compared to RCC without sarcomatoid features; thus, its description needs to be included in the surgical pathology report [3]. To which extent this would be necessary in tumors showing only 1% of sarcomatoid features is questionable, since pathology reports are extremely dependent on gross sampling.

Genomic profiling on paired epithelial and sarcomatoid areas of sRCC by next-generation sequencing confirmed different driver mutations between sRCC and ccRCC. However, the epithelial and sarcomatoid components of sRCC showed identical mutational profiles, with *TP53* (42%), *VHL* (35%), *CDKN2A* (27%), and *NF2* (19%) being the most frequently altered genes [12]. These findings have been confirmed by Wang et al. [13]: The epithelioid and sarcomatoid components of sRCC did not show differences in mutational load amongst cancer-related genes, whereas sRCC had a completely different molecular pathogenesis and distinctive mutational and transcriptional profiles compared to ccRCC. Indeed, the authors found fewer deletions at 3p21-25, a lower rate of two-hit loss of *VHL* and *PBRM1*, but more mutations in *TP53*, *PTEN*, and *RELN* [13]. Moreover, mutations in known cancer drivers, such as AT-rich interaction domain 1A (ARID1A) and BRCA1 associated protein 1 (BAP1), were significantly mutated in sarcomatoid patterns and mutually exclusive with TP53 and each other [14]. These data corroborate the hypothesis that both epithelial and sarcomatoid components of RCC may originate from the same progenitor cell, but clonal divergence occurs during tumor progression. This implicates that specific genes are involved in this process, leading to unique genetic alterations based on the observed EMT [1,15].

Induction of EMT may upregulate the expression of PD-L1 and other targetable immune checkpoint molecules in various cancer entities, such as claudin-low breast cancer [16], non-small cell lung cancer [17], or RCC [18] in vivo and in vitro. Interestingly, sRCC has been shown to express programmed death 1 (PD-1) and its ligand (PD-L1) at a much higher level than RCC without sarcomatoid elements [19], as seen in Figure 1 and Figure 2. As higher tumoral PD-L1 expression seems to correlate with higher Fuhrmann grade [20], it is essential to compare the PD-L1 status between sRCC and grade 4 non-sarcomatoid ccRCC specifically [19]. Although sRCC is defined as grade 4 RCC, tumoral PD-L1 expression in the epithelioid component of sRCC was even higher than in non-sarcomatoid grade 4 ccRCC [19]. These results may suggest a biologic distinctiveness of sRCC compared to non-sarcomatoid ccRCC at the level of immune markers [19]. In this regard, tumoral PD-L1 and PD-1 expression was found in 54% and 96% of sRCC, compared to 17% and 62% of ccRCC specimens [21]. Moreover, the co-expression of both PD-L1 on tumor cells and PD-1 positive tumor-infiltrating lymphocytes was confirmed in 50% of all sRCC cases, compared to only one case (3%) with ccRCC [21]. These findings suggest that blockade of the PD-1/PD-L1 axis could be an attractive therapeutic approach in EMT-derived tumors, such as sRCC.

A small retrospective study by Ross et al., 2018, on response to checkpoint inhibitors in RCC patients with sarcomatoid differentiation presented as an abstract at American Society of Clinical Oncology (ASCO) Annual Meeting 2018 showed promising outcomes, with durable complete responses (CR) in up to 15% of patients, and an objective response rate (ORR) of 62% [22]. The genomic biomarker analyses of the phase III IMmotion151 study on bevacizumab plus atezolizumab versus sunitinib correlated angiogenesis and immune gene expression signatures with clinical outcomes from 832 RCC patients, focusing on sarcomatoid histology. Interestingly, the PD-L1 prevalence was higher in sRCC (63%), compared to non-sRCC (39%), whereas angiogenesis gene signature was lower in sRCC (34% vs. 65%) [23]. These results may explain why sRCC patients (*n* = 86) in the PD-L1+ study group showed the greatest therapeutic benefit with atezolizumab plus bevacizumab (progression-free survival: HR, 0.56 (95% CI: 0.38–0.83)) compared to sunitinib monotherapy [23,24]. In addition, the retrospective subgroup analysis of 112 sRCC intermediate or poor-risk patients from the CheckMate214 study [25] confirmed a higher rate of PD-L1 expression (≥1%) in sRCC than in non-sRCC (47–53% vs. 26–29%). More importantly, immunotherapy with nivolumab plus ipilimumab achieved an unprecedently high ORR of 57%, with a CR rate of 18% and a median overall Survival (OS) of 31 months compared to vascular endothelial growth factor (VEGF)-targeted monotherapy with sunitinib (ORR, CR, median OS: 19%, 0% and 14 months) [26].

## 3. Conclusions

In summary, sRCC shows the typical features of EMT. On the molecular level, transcriptional data confirmed that sRCC is not a homogeneous RCC subtype and segregates primarily according to the underlying parental epithelial subtype (ccRCC, papillary or chromophobe RCC). Current biomarker studies have shown that sRCC tumors express PD-1/PD-L1 at a much higher level than non-sarcomatoid RCC. These findings will ultimately lead to a change in the treatment paradigm, shifting therapeutic decisions towards checkpoint inhibitors as first line treatment for sRCC.

## Figures and Tables

**Figure 1 cancers-11-00422-f001:**
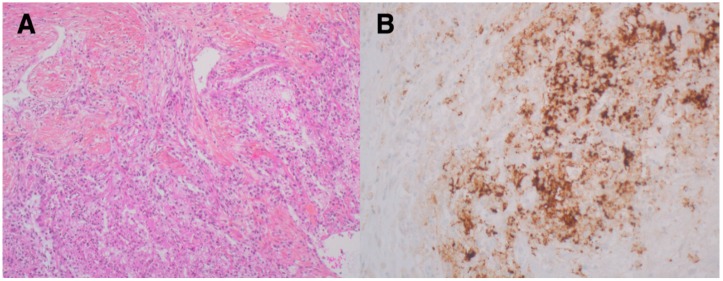
(**A**) Hematoxylin/eosin and phloxin staining for clear-cell RCC (ccRCC) with sarcomatoid features (50× magnification), considered as grade 4 according to the International Society of Urologic Pathologists (ISUP). (**B**) High PD-L1 (100× magnification) (E1L3N XP Rabbit mAB) cytoplasmatic staining of tumor cells in sRCC (>50% of tumor cells).

**Figure 2 cancers-11-00422-f002:**
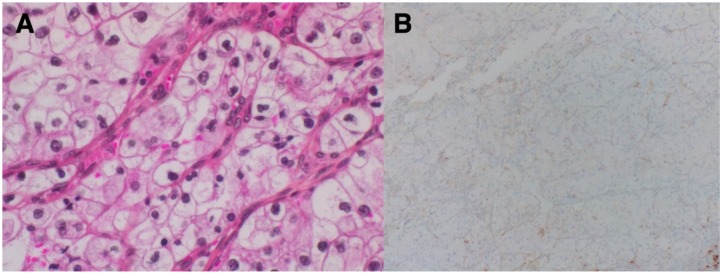
(**A**) Hematoxylin/eosin and phloxin staining for ccRCC without sarcomatoid features (400× magnification). (**B**) PD-L1 (E1L3N XP Rabbit mAB) cytoplasmatic staining of tumor cells, showing low PD-L1 expression (<5%), 100× magnification.

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
