# Peer review of "Renal Cell Carcinoma with Sarcomatoid Features: Finally New Therapeutic Hope?"

_cancers, 2019, doi:10.3390/cancers11030422_

Reviewer 1 Report

Pichler et al. reviewed recent studies on sarcomatoid renal cell carcinoma, especially its genomic profiling, association with epithelial-mesenchymal transition, immune checkpoint molecules and therapeutic effect.  I think this commentary is very interesting.  However, this paper would become much better if following points are considered.

Minor criticisms:

#1. PD-L1 is expressed both on tumor cells and tumor infiltrating immune cells.  Which is more relevant to the therapeutic effect?

#2. Morphologically, sarcomatoid change can be considered as part of nuclear grade.  It is also known that increased PD-L1 and PD-1 expression are observed in high grade renal cell carcinoma compared to low grade tumors. 

Author Response

Pichler et al. reviewed recent studies on sarcomatoid renal cell carcinoma, especially its genomic profiling, association with epithelial-mesenchymal transition, immune checkpoint molecules and therapeutic effect.  I think this commentary is very interesting.  However, this paper would become much better if following points are considered.

Minor criticisms:

#1. PD-L1 is expressed both on tumor cells and tumor infiltrating immune cells.  Which is more relevant to the therapeutic effect?

Answer: Thanks for your concerns. Most trials concerning PD-L1 expression on RCC focused on the PD-L1 status on tumor cells. This was now added more in detail in the manuscript. We added: "In this regard, tumoral PD-L1 and PD-1 expression was found in 54% and 96% of sRCC, compared to 17% and 62% of ccRCC specimens [21]. Moreover, the co-expression of both PD-L1 on tumor cells and PD-1 positive tumor-infiltrating lymphocytes was confirmed...."

#2. Morphologically, sarcomatoid change can be considered as part of nuclear grade.  It is also known that increased PD-L1 and PD-1 expression are observed in high grade renal cell carcinoma compared to low grade tumors. 

Answer: We agree totally with you. A lot of trials have shown that the PD-L1 expression correlates with aggressive histologic Features such as Fuhrmann grade in RCC. We added now in the manuscript the following sentence: "As higher tumoral PD-L1 expression seems to correlate with higher Fuhrmann grade [20], it is essential to compare the PD-L1 status between sRCC and grade 4 non-sarcomatoid ccRCC specifically [19]. Although sRCC is defined as grade 4 RCC, tumoral PD-L1 expression in the epithelioid component of sRCC was even higher than in non-sarcomatoid grade 4 ccRCC [19]. These results may suggest a biologic distinctiveness of sRCC compared to non-sarcomatoid ccRCC at the level of immune markers [19]."

Reviewer 2 Report

Well written commentary describing the challenges and future directions for treating sarcomatoid RCC.

Major Points:

- None

Minor Points:

- Consider specifying when you are citing a meeting abstract vs. a peer-reviewed publication.  I believe citation 21 might only be an abstract.

- Spelling error for "versus" in line 86

Author Response

Well written commentary describing the challenges and future directions for treating sarcomatoid RCC.

Major Points:

- None

Minor Points:

- Consider specifying when you are citing a meeting abstract vs. a peer-reviewed publication.  I believe citation 21 might only be an abstract.

Answer: Thank you very much for your comment. We added now the following sentence: "A small retrospective study by Ross et al (2018) on response to checkpoint inhibitors in RCC patients with sarcomatoid differentiation presented as abstract at ASCO 2018 showed promising outcomes, with durable complete responses (CR) in.................."

- Spelling error for "versus" in line 86

Answer: Thanks, this spelling error was now corrected.

Reviewer 3 Report

The manuscript describes renal cell carcinoma (RCC) under the spotlight of sarcomatoid RCC (sRCC) versus non-sarcomatoid RCC with regard to biomarker studies. Both RCC subtypes differ in PD-1/PD-L1 expression which the authors claim to ultimately lead to change in treatment paradigm towards so called checkpoint inhibitors in sRCC patients

Overall the comment in its shortness is well written and can be published with these minor comments in mind:

Spelling mistake line 86 "versus".

Citation style is different from the guidelines.

Figures should include a size bar.

Figure legends line 99 Phloxin or line 103 phloxin inconsistent spelling.

Figure 1 versus 2: It would be better if both figures would display the same magnifications for better comparison. What is the rationale behind every picture displays a different magnification?

For such a short commentary the list of authors seems a bit long especially how many people were needed for "critical revision".

Author Response 

The manuscript describes renal cell carcinoma (RCC) under the spotlight of sarcomatoid RCC (sRCC) versus non-sarcomatoid RCC with regard to biomarker studies. Both RCC subtypes differ in PD-1/PD-L1 expression which the authors claim to ultimately lead to change in treatment paradigm towards so called checkpoint inhibitors in sRCC patients

Overall the comment in its shortness is well written and can be published with these minor comments in mind:

Spelling mistake line 86 "versus".

Answer: Thanks, this was already corrected.

Citation style is different from the guidelines.

Answer: Thank you so much. The citation style was now adapted according to the guidelines.

Figures may should include a size bar.

Answer: Thank you very much for your concerns. As we already included the exact magnification of each pathological figure, we think that this graphical specification should be sufficient for publication.  

Figure legends line 99 Phloxin or line 103 phloxin inconsistent spelling.

Answer: Thanks for this comment, it was now corrected consistently to "phloxin staining".

Figure 1 versus 2: It would be better if both figures would display the same magnifications for better comparison. What is the rationale behind every picture displays a different magnification?

Answer: Thank you for this comment. The reason of the different magnifications was that on figure 1 we wanted to focus more on the high PD-L1 expression (therefore x100) and on figure 2, we wanted to show more in detail the HE staining of a non-sarcomatoid ccRCC, whereas the magnification of x100 was chosen for PD-L1 as the expression was very low to have a better overview.      

For such a short commentary the list of authors seems a bit long especially how many people were needed for "critical revision".

Answer: I agree with you. The reason why we are in total 7 authors is the fact that we all present an Austrian RCC expert panel and to write this comment on sRCC was the idea of our last RCC meeting. So this is the reason why we are 7 authors in total.